# A method to estimate population densities and electricity consumption from mobile phone data in developing countries

**Hadrien Salat** [1,2]*, **Zbigniew Smoreda**[2], **Markus Schläpfer**[1]

**1** Future Cities Laboratory, Singapore-ETH Centre, ETH Zürich, Singapore, Singapore, **2** Sociology and Economics of Networks and Services department, Orange Labs, Châtillon, France

* hadrien.salat@orange.com

## Abstract

High quality census data are not always available in developing countries. Instead, mobile phone data are becoming a popular proxy to evaluate the density, activity and social characteristics of a population. They offer additional advantages: they are updated in real-time, include mobility information and record visitors' activity. However, we show with the example of Senegal that the direct correlation between the average phone activity and both the population density and the nighttime lights intensity may be insufficiently high to provide an accurate representation of the situation. There are reasons to expect this, such as the heterogeneity of the market share or the particular granularity of the distribution of cell towers. In contrast, we present a method based on the daily, weekly and yearly phone activity curves and on the network characteristics of the mobile phone data, that allows to estimate more accurately such information without compromising people's privacy. This information can be vital for development and infrastructure planning. In particular, this method could help to reduce significantly the logistic costs of data collection in the particularly budget-constrained context of developing countries.

## Introduction

Mobile phone data allow, under certain conditions, to recover a map of the population and can potentially simplify the logistics of census data collection [1–4]. This could prove particularly useful in developing countries where such costs cannot be overlooked. However, these approaches have only been validated in developed countries where detailed fine-grained data are comparatively easier to access to train the models. Furthermore, a primary objective of population mapping is to inform infrastructure planning. In that respect, mobile phone data have a number of advantages over a simple population count. They represent some notion of intensity of activity, include dynamic real-time usage information and contain mobility patterns. For example, significant results have been obtained for the prediction of short-term population dynamics inside cities [5, 6] and for the prediction of detailed socioeconomic characteristics of users from metadata [7–9]. However, these methods once again require a large

**Data Availability Statement:** The census data for the year 2013 in Senegal and the nighttime lights data can be accessed directly through the public databases of ANSD and NOAA (links provided in the article). The mobile phone data at Voronoi and Commune levels and aggregated over the year are

available as part of the supplementary material. The identity of the callers has been removed and the exact location of the communication towers has been slightly modified for confidentiality reasons. In addition, a time series containing the number of calls per hour for the month of January is also available as part of the supplementary material. To obtain the dataset over the entire year, one would need to contact Sonatel directly and present the research project that would require the data (contact: Mr El Hadji Birahim Gueye, Direction des Systèmes d'information Sonatel, ebgueye@orange-sonatel.com or post mail: Orange-Sonatel, 46 Boulevard de la République, BP 69 Dakar, Senegal).

**Funding:** H.S. was supported by the Orange Labs-Sonatel-ETH Singapore SEC Research Agreement No. H11283. M.S. acknowledges the Future Cities Laboratory at the Singapore-ETH Centre, which was established collaboratively between ETH Zurich and Singapore's National Research Foundation (FI 370074016) under its Campus for Research Excellence and Technological Enterprise Programme.

**Competing interests:** The authors have declared that no competing interests exist.

amount of fine-grained data, up to the individual level and its associated privacy concerns, to train the models or may require additional sources of data such as satellite images.

Building on these pioneering studies, we propose a new method, based on the daily, weekly and yearly phone activity curves and on the network characteristics of the mobile phone data (such as node centrality or incoming to outgoing traffic ratio), that is focused on the specific context of data scarcity in developing countries and that does not require information about identified individuals. Some earlier work has revealed mobile phone data's particular potential for predicting electricity demand [10, 11]. In sub-Saharan Africa, electrification rates remain extremely low, without much optimism for a rapid improvement of the situation [12–15]. In 2013 in Senegal, when the last census was collected, the average electrification rate in rural areas was as low as 24%. Paradoxically, mobile phones have still found their way into the homes of about 75% of the population in these same rural areas. In fact, some studies praise the large coverage achieved by mobile phones in the entire African region [16, 17]. Our aim is to use the resulting data to guide efficient manual data collection and therefore reduce the logistic costs of gathering information for development and infrastructure planning in developing countries. We both test the possibility of estimating census data from mobile phone data and evaluate the potential of better predicting electricity demand from mobile phone data rather than from the population count. We first describe our method in details in the materials and methods section, then use a bulk of data that we have gathered for Senegal in 2013, including the aggregated call detail records from Sonatel, the leading mobile phone operator in Senegal (with 65% market share), to validate our proposed approach.

## Materials and methods

The population density for each commune in Senegal is given by the 2013 census. There are 552 communes of irregular sizes according to the division provided (created in December 2013), including urban communes (*communes de ville* and *communes d'arrondissement*) and rural communes (*communautés rurales*). This information was collected using a door to door approach over the entire country, rather than by estimation. The population densities' distribution is close to a narrow Poisson distribution, with an average of 2162 inh./km$^2$ and a maximum of 54325 inh./km$^2$. The distribution is mapped over a rather precise shapefile that includes in particular the boundaries of all big and medium towns. We use mobile phone data provided by the largest Senegalese telecommunication operator, with about 65% market share in 2013, Sonatel. They contain the number of text messages, number of calls and total length of calls made during each hour between each of the operator's 1666 communication towers during the year 2013. Out of these, 54 towers were inactive and have been removed from the analysis, leaving 1612 towers. To estimate the electricity consumption, we used NOAA's average nighttime lights intensity for the year 2013. The intensity is given as a number between 0 and 63 for each cell of a 30 second arc grid. Since Senegal is close to the equator, this grid is regular and its cells measure about 1km per 1km. This data has been cleaned by NOAA from the interference of moonlight, clouds, etc. to the best of their ability.

We have produced two different levels of aggregation. The first one is the commune level. It is aimed at preserving the population counts as accurate as possible. The second consists of Voronoi cells around each tower. It aims at preserving the mobile phone data as precise as possible. In this case, the population count inside each Voronoi cell has been estimated from the intersection between the Voronoi cell and the communes, assuming a uniform distribution inside each commune. Since the small high and medium density communes are finely separated from the big low density communes in the geographical data, this assumption seems reasonable. The nighttime light intensity is averaged inside each Voronoi cell or commune. If a

pixel is only partially included in the cell, its intensity is weighted in the average by the area actually included in the cell. The end result is two tables containing the population count per square kilometre, the average number of texts, average number of calls and average total call length per hour per square kilometre, and the nighttime light intensity per pixel, inside each of the 552 communes and each of the 1612 Voronoi cells.

For the record, we also computed the total value of each variable inside the cells instead of their density per square kilometre. We found consistently better correlations between densities rather than between total values. The results for the total values are therefore not shown in this paper. In addition, since nighttime lights and residential locations represent a better picture of the night activity rather than the day activity, we isolated the texts and calls made between 7 p.m. and 7 a.m. Finally, the communes have been further divided into 444 "low density" communes and 108 "high density" communes, corresponding to a density lower or higher than 1000 inhabitants per pixel, to distinguish between mainly rural and mixed or purely urban areas. This threshold is arbitrary, but commonly used, for example by the Food and Agriculture Organization (FAO) of the United Nations and the Global Rural Urban Mapping Project (GRUMP). Similarly, the Voronoi cells have been divided into 1027 "low density" cells and 639 "high density cells" according to the same criteria.

The direct correlations shown in the results are simple squared Pearson coefficients. To exploit further the data, we compute some curves representing for each hour of the day, week and year, the average number of texts, number of calls and total call length for each tower site. These 9 types of curves represent the local phone usage profiles. They are used to generate matrices, called *distance matrices*, based on the point-by-point correlation and the standard deviation between the point-by-point distances between two curves to evaluate how point-by-point "parallel" they are. Here, each point represents each hour contained in a day, week or year and "point-by-point" means that we are comparing the values between the curves at each hour. This gives a total of 18 distance matrices. We also exploit the characteristics of the data's network structure. We transform it into weighted directed graphs averaged over the entire year. An edge is created between two cell towers if the daily activity is above a predefined threshold. We use five thresholds: 0, 75, 150, 300 and 600, based on the histogram of the activity between two pair of nodes (approximately an inverse power law of exponent 1.81). We then create feature matrices recording for all nodes their degree, betweenness and closeness centrality measures (both weighted and unweighted), the ratio of self-loops to the total traffic, the ratio between the number of incoming and outgoing traffic and the average distance travelled by a text message or call. With this process, we obtain an additional 15 feature matrices.

As a reference, the hourly curves for the number of calls aggregated at national level for each day of the year are represented in Fig 1(a). There is one colour per month ranging from reds to yellows to greens to blues. The yearly average of number of texts per hour of the day sent from each tower is shown in random colours in Fig 1(b). The network structure in January limited to edges corresponding to at least 2000 text messages sent is represented in Fig 1(c).

Our proposed method consists in trying to rebuild the original dataset from a sample as small as possible using hierarchical clustering of some of the 33 previously created matrices. The working hypothesis is that similar locations will share in particular similar phone activity habits and have a similar place in the communication network. In the first step, we build a dendrogram from the distance and feature matrices using the *hclust* hierarchical clustering algorithm implemented in *R*. Assuming that we know the population density or nighttime lights intensity for a number of reference towers, the values of all the other towers are predicted from the proximity of their activity curve or network characteristics to the activity curves or network characteristics of the reference towers. Specifically, the value for a non-reference tower is set to

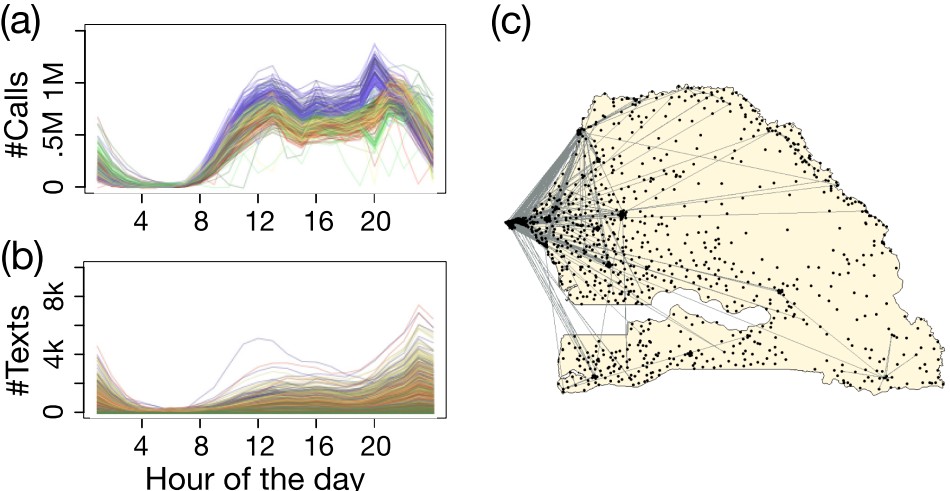

**Fig 1. Mobile phone activity profiles and network structure.** (a) Number of calls per hour aggregated at national level for each day of the year. (b) Yearly average of the number of texts per hour of the day sent from each tower. (c) Network structure limited to edges corresponding to at least 2000 text messages sent in January.

be equal to that of the closest reference tower for the chosen distance. We compare the performance of samples of reference towers chosen randomly with samples chosen according to the clustering tree.

To choose a sample according to the clustering tree, we first cut the tree at a chosen depth, then generate the sample by including one randomly chosen leave per resulting branch. By construction, when the selected depth is increased, the sample size is also increased. In particular, the depth can be chosen to match a desired sample size. To illustrate this process, consider the tree corresponding to the daily number of texts used above. It is shown in Fig 2(a). Each branch is identified by a binary number counting the number of left turns (indicated by a 0) and right turns (indicated by a 1) that are necessary to reach it while scrolling the tree starting from the top. Five illustrative clusters, evidenced by a colour code, are plotted over a map of Senegal in Fig 2(b). We can observe that the blue cluster identifies mostly rural areas, the orange one is mixed, while the other clusters identify only cities. For a chosen depth, equivalent to a binary numbers' length, we select one random leave in each induced cluster to populate the sample. Examples are given in Fig 2(c). With a depth of 3 (dark blue), we obtain 0.5% of all towers, with a depth of 7 (medium blue), we obtain 3.3% of all towers, and with a depth of 19 (teal), 44.4% of all towers. Naturally, in the event of a branch being reduced to only one leave before the chosen depth is reached, this leave is kept in the sample and the branch is not divided further. There may therefore be fewer elements in the sample than the power of two of the chosen depth.

The census data for the year 2013 in Senegal can be directly accessed through the official website. The nighttime lights data can be accessed through NOAA's open database. The mobile phone data at Voronoi and Commune level and aggregated over the year are available as part of the supplementary material (S1 and S2 Files). The identity of the callers has been removed and the exact location of the communication towers has been slightly modified for confidentiality reasons. In addition, a time series containing the number of calls per hour for the month of January is also available as part of the supplementary material (S3 File). To obtain the dataset over the entire year, one would need to contact Sonatel directly and present the research project that would require the data (contact: Mr El Hadji Birahim Gueye, Direction des

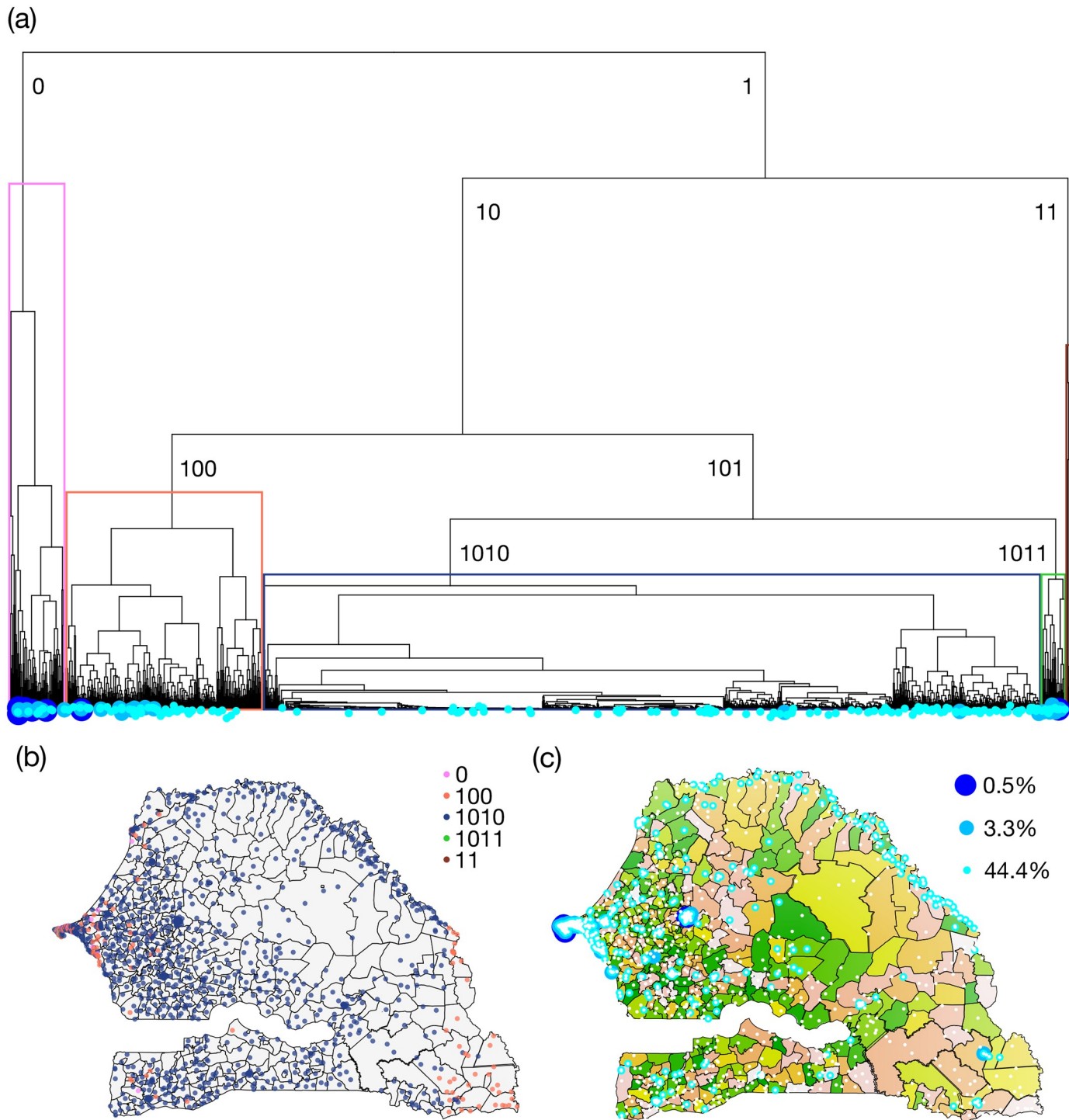

**Fig 2. Description of the tree cutting process.** (a) Tree corresponding to the daily number of texts. The branches are identified by a binary number. (b) Locations of the members of five clusters identified on the tree. (c) Locations of one randomly chosen element per branch of the tree cut at depth 3 (dark blue), 7 (medium blue) and 19 (cyan).

Systèmes d'information Sonatel, ebgueye@orange-sonatel.com or post mail: Orange-Sonatel, 46 Boulevard de la République, BP 69 Dakar, Senegal). The analysis was performed using *R*.

This research uses data from Senegal. It was approved by the Senegalese *Commission de Protection des Données Personnelles* (private data protection commission) on the 13th of July 2015 as part of the "Data for Development (D4D)" project. This analysis only uses data that was de-identified by the mobile phone operator before accession by the authors.

## Results and discussion

In Table 1, the squared Pearson correlation coefficients ($r^2$) of population density and night-time lights versus the average daily number of texts, number of calls and total call length per square kilometre at tower and commune levels are reported. The average hourly values of the number of text messages and calls and of the total call length per tower are only moderately correlated with the local population density at tower level in Senegal, with $r^2$ values of 0.4-0.6. The results are better at the much coarser commune level, with values between 0.6-0.8. The mobile phone variables are also moderately correlated with the nighttime light intensity with values around 0.4 at tower level and 0.65 at commune level. Reducing the mobile phone activity to nighttime activity does improve the correlations for calls, but not for text messages.

There are a number of reasons to explain why we could expect these relatively low values. Although Sonatel, the operator curating the mobile phone data, is the market leader, its market coverage is not uniform over the entire country. In addition, the only population data that was available remains at a relatively coarse spatial resolution. Unfortunately, these limitations cannot be avoided in the context of Senegal in 2013. Another limitation is the lack of precision of the Voronoi modelling that does not take into account congestion and hand offs among the towers.

We now show how we can significantly improve these predictions with our method. We focus on the tower level, as this aggregation level is far more interesting for planning endeavours than the coarse "commune level". Fig 3(a) reports the $r^2$ values of the population density

**Table 1. Population density (Pop.) and nighttime lights (Elec.) correlations ($r^2$) with number of texts, calls and total call length daily average values (per km$^2$) at tower and commune levels.** (n) means that only the activity between 7 p.m. and 7 a.m. was included. For each case, three results are given: all areas included, only low density areas included and only high density areas included. All p-values are $< e^{-15}$.

**Voronoi cells around towers**

|  | Texts | Calls | Length | Texts (n) | Calls (n) | Length (n) |
|---|---|---|---|---|---|---|
| Pop. all | 0.43 | 0.45 | 0.46 | 0.47 | 0.62 | 0.60 |
| Pop. low density | 0.26 | 0.25 | 0.26 | 0.27 | 0.23 | 0.24 |
| Pop. high density | 0.24 | 0.25 | 0.26 | 0.29 | 0.46 | 0.43 |
| Elec. all | 0.39 | 0.41 | 0.44 | 0.39 | 0.43 | 0.45 |
| Elec. low density | 0.64 | 0.33 | 0.37 | 0.60 | 0.27 | 0.30 |
| Elec. high density | 0.20 | 0.19 | 0.21 | 0.20 | 0.21 | 0.23 |

**Communes**

|  | Texts | Calls | Length | Texts (n) | Calls (n) | Length (n) |
|---|---|---|---|---|---|---|
| Pop. all | 0.59 | 0.73 | 0.72 | 0.61 | 0.81 | 0.79 |
| Pop. low density | 0.85 | 0.93 | 0.92 | 0.84 | 0.92 | 0.91 |
| Pop. high density | 0.50 | 0.67 | 0.65 | 0.53 | 0.76 | 0.73 |
| Elec. all | 0.59 | 0.66 | 0.66 | 0.58 | 0.65 | 0.66 |
| Elec. low density | 0.59 | 0.65 | 0.65 | 0.58 | 0.62 | 0.62 |
| Elec. high density | 0.53 | 0.61 | 0.61 | 0.53 | 0.60 | 0.61 |

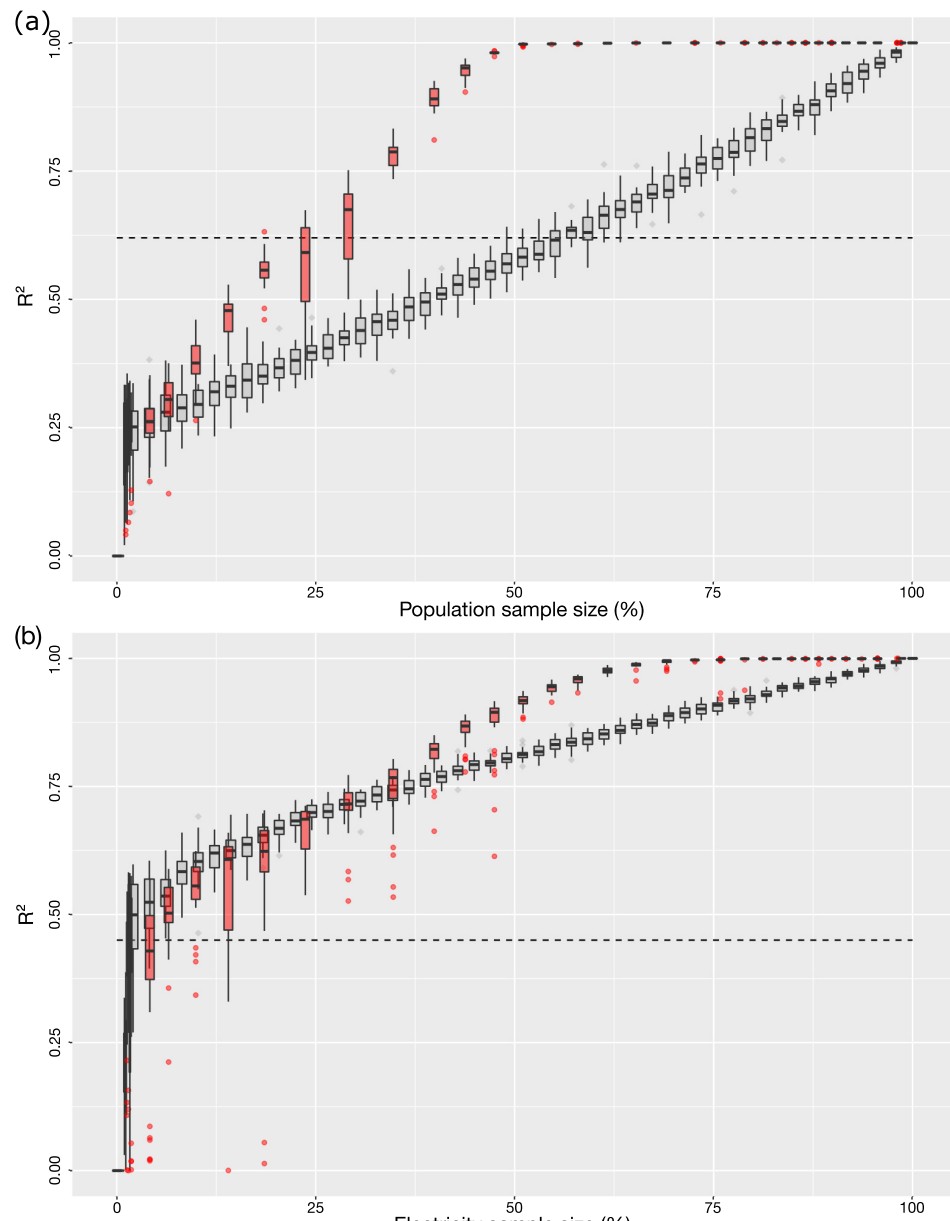

**Fig 3. Recovering population density and nighttime lights from a sample of reference mobile phone activity curves.** (a) Population density. (b) Nighttime lights intensity. The grey boxes are for a sample randomly chosen among all curves. The red boxes are for a sample selection guided by the daily number of texts profile clustering tree. The best direct correlation from Table 1 for each case is represented as a horizontal dash lines. All four cases have been tested 30 times to build the boxes (lower and upper quartiles) and whiskers.

distribution predicted from samples (expressed as a percentage of the entire distribution) guided by the dendrogram produced from the daily texts activity curves' standard deviation. It is plotted using black dots against predictions from fully random samples of increasing size in grey squares. The boxes (lower and upper quartiles) and whiskers have been computed experimentally by repeating the process 30 times. Fig 3(b) repeats the process with electricity distributions. The maximum $r^2$ value achieved for each case by the direct correlations is indicated as a horizontal dash line. Additional figures based on samples guided by some of the other best

performing clustering trees are provided in the supplementary material (S1 Fig of S1 Appendix). A table comparing the performance of each of the 33 prepared clustering matrices is then shown in S1 Table of S1 Appendix. Only the curve-based clustering is shown here since it consistently provided slightly better results than the network-based clustering for the particular phone usage habits in Senegal.

We can immediately observe that using the tree as a guide has a major impact on the quality of the results, especially for population density predictions. For example, in panel (a), the technique outperforms direct density correlations with samples as small as 30%, and allows obtaining an $r^2$ of almost 1 for samples as small as 55%. The difference between the tree guided sampling and the random sampling is smaller in panel (b), although we can notice that using the curves rather than the average values to identify similarities outperforms significantly direct correlations even with fully random sampling of reference curves. Finally, note that the direct squared correlation between electricity and population is only 0.51. We can therefore obtain better results from the mobile phone activity than from the population density.

To get some insights about the hidden functioning of the clustering, we show the average number of text messages sent per hour normalised by the total volume over the day in Fig 4(a) for a 4 clusters partition of the daily text messages standard deviation tree. Panel (b) shows the same content normalised by the phone traffic at 2 pm. We observe two effects: the green curve corresponding mostly to low density areas is more impacted than the red and blue curves during work to home travel time (4 to 7 pm) and at night. We can indeed hypothesise that the lack of electrification forces people to go to bed earlier in electricity deprived low density areas. Note that one cluster made only of five odd towers has been omitted in the figure.

Finally, potential errors introduced in the clustering by locally different habits in phone usage could be smoothed out by non-parametric methods such as kernel smoothing or weighted averages between several close-by reference towers. Alternatively, potential hidden biases specific to one type of usage (e.g. calls) that might not exist for another type of usage (e.g. texts) could be mitigated by averaging the results obtained from different distance matrices. In practice, we found that combining the results from several distance matrices did not improve the results compared to the most successful distance matrices considered alone. This averaging might still be necessary if a training set cannot be gathered to identify which are these most successful distance matrices for a different specific context. Another bias could come from a shift towards platforms such as *WhatsApp* and *Facebook Messenger*, or simply different usage among different age groups, leading to a possible under-estimation of certain demographics. However, since we are only establishing similarity of usage between towers, a

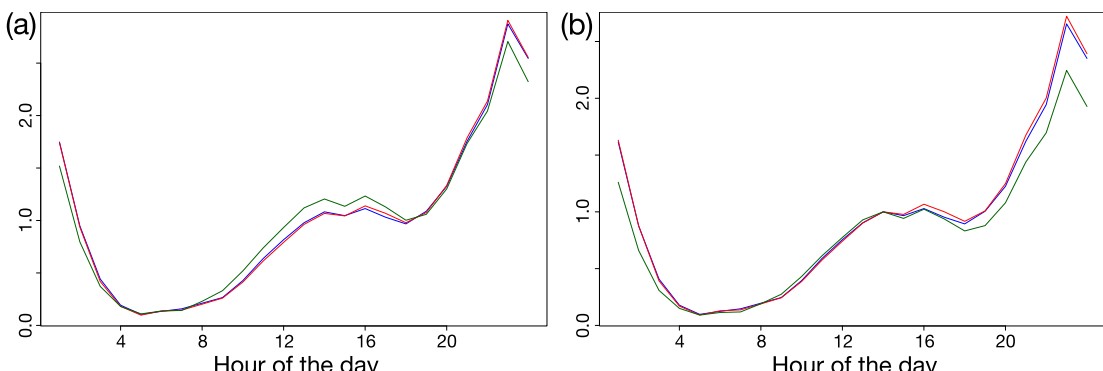

**Fig 4. Average curve inside the clusters.** (a) Average daily activity curves normalised by the overall daily volume in 3 clusters from the daily texts standard deviation dendrogram. (b) Same average activity curves normalised by the activity at 2 pm.

shift in usage that does not break similarity between places does not impact the results. In particular, smartphones and mobile data are still excessively expensive for a widespread usage in the area [17]. Internet users are therefore only going to be found in sufficiently large number in the richest areas, allowing the method to differentiate more these areas from other areas, rather than blurring the results. Similarly, if the variable 'age distribution inside an area'has a significant impact on phone usage then the clustering will group together places with similar age distributions, so that these will be represented by adequate reference towers.

## Conclusion

Our first important result is that the average mobile phone activity is not necessarily well correlated with the population density, an idea that became particularly tempting after the seminal work by Lu et al. who predicted population displacements after a natural disaster from mobile phone data [18]. With our methodology, we introduce new perspectives. We have shown that the population count of an entire census can be estimated from a substantially smaller sample of carefully selected locations with the help of the clustering trees and without requiring the Call Data Records of identified individuals. There are two main practical applications to this: a reduction in data collection costs to about one half in the best case scenarios and the possibility to keep tracking the changes in the population distribution between two census surveys. This technique could complement for example the approach by Lai et. al. which shares the same aim, but uses direct regressions from average values only [19]. Note that contrary to previous methods, the clustering relies solely on mobile phone data, without requiring a population training dataset. In addition, it can accommodate external information known *a priori*. It should be possible for example to use satellite data to subset potential reference locations into obviously low or obviously high density areas. This is a step in the direction proposed by the director of UN Global Pulse, Robert Kirkpatrick, who asserted that "the next phase in call-records research should be cost–benefit analyses that look at the investment needed to conduct a study, roll out an intervention and appraise the advantages for communities." [20].

We appreciate that our method requires access to a sufficiently large mobile phone dataset and that Senegal is a leader in sub-Saharan Africa in terms of electrification rate, mobile phone penetration and census data collection. However, since we are not using the full mobile phone penetration rate, but only the 65% market share of Sonatel, we believe that there should be enough underlying data in many other developing countries to apply the method there. Beware, however, that in some cases, using a single operator with an inhomogeneous market share might introduce some important biases (cheaper providers may be chosen more by people who consume less electricity for example).

## Supporting information

**S1 File. Mobile phone data at Voronoi level and aggregated over the year.** The table contains an id of the Voronoi cell, the longitude and latitude coordinates of each Voronoi centre (slightly modified for privacy reasons), the average population density and nighttime light intensity per $km^2$ inside the cell, and the number of text messages, calls and total call length per $km^2$ for each cell.
(ZIP)

**S2 File. Mobile phone data at Commune level and aggregated over the year.** Equivalent to the previous file, but at Commune level.
(ZIP)

**S3 File. Time series of outgoing calls for each Voronoi cell in January.** The table contains a Voronoi cell id, a time stamp for each hour of the month and the number of outgoing calls during this hour in the cell.
(ZIP)

**S1 Appendix. Additional methods and figures.** Alternative method to estimate nightlights intensity from approximate data, validation of the performance of the clustering process and additional figures.
(PDF)

## Acknowledgments

The authors acknowledge Aike Steentoft for his guidance in choosing an adequate methodology to cluster the mobile phone activity curves.

## Author Contributions

**Conceptualization:** Hadrien Salat, Zbigniew Smoreda, Markus Schläpfer.

**Data curation:** Zbigniew Smoreda.

**Formal analysis:** Hadrien Salat.

**Methodology:** Hadrien Salat, Markus Schläpfer.

**Writing – original draft:** Hadrien Salat.

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
