## [Decision Letter · Decision Letter 0]

14 Apr 2020

PONE-D-20-02661

A method to estimate population densities and electricity consumption from mobile phone data in developing countries

PLOS ONE

Dear Dr. Salat,

Thank you for submitting your manuscript to PLOS ONE. After careful consideration, we feel that it has merit but does not fully meet PLOS ONE’s publication criteria as it currently stands. Therefore, we invite you to submit a revised version of the manuscript that addresses the points raised during the review process.

Both reviewers liked your paper but also raised some conerns for revision. Please address their concerns as much as you can in the revision. I myself also have a question. As a careful reader I want to know more about how you make the prediction. Specifically, you wrote in lines 105-107  "the values of all the other towers are predicted from the proximity of their activity curve or network characteristics to the activity curves or network characteristics of the reference towers." How is the prediction exactly implemented? Do you use some nonparametric method such as kernal smoothing or weighted average? I hope you can provide more details on how you generate the prediction.

We would appreciate receiving your revised manuscript by May 29 2020 11:59PM. To enhance the reproducibility of your results, we recommend that if applicable you deposit your laboratory protocols in protocols.io, where a protocol can be assigned its own identifier (DOI) such that it can be cited independently in the future. For instructions see: http://journals.plos.org/plosone/s/submission-guidelines#loc-laboratory-protocols

We look forward to receiving your revised manuscript.

Kind regards,

Shihe Fu, Ph.D.

Academic Editor

PLOS ONE

Journal Requirements:

Reviewers' comments:

Reviewer's Responses to Questions

**Comments to the Author**

1. Is the manuscript technically sound, and do the data support the conclusions?

Reviewer #1: Partly

Reviewer #2: Yes

2. Has the statistical analysis been performed appropriately and rigorously? 

Reviewer #1: No

Reviewer #2: Yes

3. Have the authors made all data underlying the findings in their manuscript fully available?

Reviewer #1: Yes

Reviewer #2: No

4. Is the manuscript presented in an intelligible fashion and written in standard English?

Reviewer #1: Yes

Reviewer #2: Yes

5. Review Comments to the Author

Reviewer #1: The paper presents a method of estimating population density and electricity consumption using mobile phone data usage data, including SMS, call and data usage. Usage characteristics of cell towers are used to generate feature matrices, which are then

used to generate a graph representation of cell towers, with similar towers having connecting edges. Network analysis is then applied to the resulting graph in order to extract additional feature matrices for degree, betweenness and closeness. This

results in several feature matrices, each describing different pairwise similarity measures of the cell towers. Hierarchical clustering is applied to towers using the computed features. Subsequently, tree cutting of the resulting dendrogram is

performed at various depths, with a leaf from each of the resulting branches then being randomly selected.

Baseline R2 measures are generated by correlating SMS/Call/data volumes with population/electricity usage data. It was found that selecting nodes using the tree cutting process could result in higher correlation scores for both population density and electricity consumption measures compared to the baseline method, with results varying according to the resulting sample size for a given tree cut depth.

To the best of my knowledge, this method, in particular applying tree cutting to the dendrogram as a means to sample cell towers, is a novel approach to the problem of population and electricity usage estimation, with results that appear promising.

The authors draw on existing peer reviewed work when formulating their method, and build on existing and respected work. However, there is a lack of references when describing the exact methods employed during the analysis, which is reflected in the

relatively short citation count.

Although the results presented in this paper appear promising, there are some areas I would like to see expanded/improved on. Specifically:

* It was not entirely clear if this method was being proposed as an *alternative* to manual data collection through census, or as a way to guide efficient collection. It may be worth clarifying this point.

* SMS (and to a lesser extent, traditional mobile phone calls) volumes are decreasing in many countries (https://www.statista.com/statistics/271561/number-of-sent-sms-messages-in-the-united-kingdom-uk/), with shifts towards platforms such as WhatsApp and Facebook Messenger. If this is the case in Senegal, then the model is likely to be less effective in 2020 compared to 2013, and may result in the under-estimation of certain demographics (if, for instance, younger people are more likely to use alternatives to SMS). It may be worth addressing this point.

* P-Values are not presented in the evaluation of either the baseline or the proposed model. It would be good to see these, if possible.

* Although error bars are presented (by running the model 30 times with different random seeds), I would be interested in seeing more analysis around the sensitivity wrt. the random selection process.

* Some of the constants chosen appear fairly arbitrary; for instance the five thresholds mentioned on L87 and the 1,000 inhabitant threshold mentioned on L73. Consider explaining how they were chosen.

* A brief discussion on the type of data collected within the Senegal census may be relevant here. The authors claim that "an entire census can be estimated", however only population density and electricity consumption levels are estimated. This may be because the Senegalese census consists exclusively of population count, but this should be explained.

* Several feature descriptors are used for the hierarchical clustering process -- however, these features are not directly used when modelling population density/electricity consumption. This (superficially) seems like a wasted opportunity, it may be worth explaining why?

* A minor point, but the X-ticks for hour of day plots may be slightly more natural as [4, 8, 12, 16, 20]?

Overall, this work has strong potential, but in my opinion requires some additional work before publication.

Reviewer #2: Summary:

"A Method to estimate population densities and electricity consumption from mobile phone data in developing countries" provides a good method to evaluate the population density and activity (electrical usage) based on the mobile phone data. Their proposed method is utilizing a machine learning algorithm – hierarchical clustering to recover an entire census from a very small sample in the census using daily, weekly and yearly mobile activity. The selection of this small sample is based on the clustering algorithm. Since mobile data is relatively easier to get than the high-quality census data, this method has a high potential to utilize mobile data to help construct the census data and at the same time, greatly reduce the costs of census data collection (by utilizing the available mobile data to impute the entire census data, the collecting cost of census data is shrinking to a much smaller training sample). In general, I think it is a very interesting paper and I have much enjoyed reading it.

General comments:

My general comment is regarding mobile phone penetration and the frequency of usage of mobile phone service in developing countries. First of all, I just googled “the percentage of the world has a cell phone in 2019”, it shows 67 percent from statista.com. I do not know how reliable the data I found on google is. But I am interested in knowing whether this method can apply to other developing countries. However, based on the results of this paper, I speculate the mobile phone penetration rate is very high in Senegal. I appreciate the authors mention in the introduction that even in low electrification rural areas, mobile phone penetration in those areas are still 75 %. Secondly, mobile phone usage may vary across ages and/or education levels. People of different ages may have quite a different percentage of mobile usage even they all own a cell phone. In some extreme cases, children or school-age teenagers may not be encouraged to have/use a mobile phone. Then the lack of information from these categories of the population may affect the prediction power of the learning algorithm. Thirdly, regarding the representability of the mobile data from one provider. I appreciate the author uses the largest Senegalese telecommunication operator’s data, 65 percent of market share. From the results, I believe in Senegal, the other providers more or less target on similar categories of people compared to Sonatel. But in some cases (if extend this method to another developing country), different providers may target different categories of people. Some people choose to use a cheaper provider and they may choose to consume less amount of electricity, which may lead to bias in the model forecasting. The overall penetration of mobile phones and their frequency of usage in a particular developing country (like Senegal) may be introduced in the introduction. It would be interesting to know if in the case of low mobile penetration and/or high diverge in mobile phone usage in a developing country, how effective this method will be and what is the authors’ recommendation to use their method to uncover census data in the above cases. Besides those, I appreciate the authors take the consideration of tower in the data aggregation.

Minor comments:

Page 1, line 17,

What are the network characteristics you refer to?

Page 3 line 83,

What is the definition of ‘distance matrices’ in your paper? Can you also give more details on the ‘point-by-point’ correlation you refer? What is the definition of ‘point’?

Page 5 line 161

What is the total number of towers? I agree that the authors remove 54 towers with no activity throughout the year in the clustering algorithm approach.

Page 5 Table 1

In table 1, illustrations in the Voronoi cells around towers section (use average mobile data instead of the clustering algorithm), as a comparison, do you also remove the 54 inactive towers? If not, why? I speculate if the inactive towers are included, it will reduce the value of the correlation.

Figure 3

Why are the correlations values of towers - calls(n) and towers-texts or towers-length(n) and towers- texts be the only ones that are selected as the horizontal lines? I can see that calls(n) are the highest correlation value in pop.all and length(n) is the highest one in elec.all (both of them are in terms of aggregating of towers).

Supporting documents – S1_Appendix, Figure S1

It would be interesting if you can also add the performance from the “fully random samples of increasing size” (the grey lines in Figure 3).

Minor typos I found from the manuscript:

Page3 line 83,

Left quotation mark on the top left of the word “parallel”.

Page 4 Line 143

The extra word “calls” after “number of calls”.

6. PLOS authors have the option to publish the peer review history of their article (what does this mean?). If published, this will include your full peer review and any attached files.

Reviewer #1: Yes: Joseph Redfern

Reviewer #2: No

---

## [Author Response · Author response to Decision Letter 0]

4 May 2020

Dear Dr. Fu,

We want to thank the Editor and the Reviewers for their useful comments and hope that the changes detailed below will answer all their questions.

Please note that all the lines indicated in the response refer to the marked-up version.

Editor's comments

> As a careful reader I want to know more about how you make the prediction. Specifically, you wrote in lines 105-107 "the values of all the other towers are predicted from the proximity of their activity curve or network characteristics to the activity curves or network characteristics of the reference towers." How is the prediction exactly implemented? Do you use some nonparametric method such as kernal smoothing or weighted average? I hope you can provide more details on how you generate the prediction.

The value for an unknown tower is set to be equal to that of the reference tower that is closest to it for the chosen distance. This is preferred over suggested kernel smoothing or weighted averages for two reasons: first, we want to make the method as simple and accessible as possible; second, since we have a multitude of distance matrices, we can preferably smooth potential errors by averaging over different distance matrices rather than over different towers for one matrix (in the style of Table~S1). This way, potential hidden biases specific to one type of usage (e.g. calls) can be mitigated. In practice, we did not find any noticeable improvement when doing so.

We have made the following changes:

Added above precision on prediction mechanics (l.118-119).

Added averaging (kernel, different matrices\\dots) as possible extensions (l.219-228).

Reviewer #1's comments

> Although the results presented in this paper appear promising, there are some areas I would like to see expanded/improved on. Specifically:

> It was not entirely clear if this method was being proposed as an *alternative* to manual data collection through census, or as a way to guide efficient collection. It may be worth clarifying this point.

This has been clarified in the introduction (l.28).

> SMS (and to a lesser extent, traditional mobile phone calls) volumes are decreasing in many countries (https://www.statista.com/statistics/271561/number-of-sent-sms-messages-in-the-united-kingdom-uk/), with shifts towards platforms such as WhatsApp and Facebook Messenger. If this is the case in Senegal, then the model is likely to be less effective in 2020 compared to 2013, and may result in the under-estimation of certain demographics (if, for instance, younger people are more likely to use alternatives to SMS). It may be worth addressing this point.

One of the strength of the method is that we are only establishing similarity of usage between towers. As a result, a shift in usage that does not break similarity does not impact the results. In particular, smartphones are still excessively expensive for a widespread usage in the area (see reference [17] where it is reported that the cost of charging a smartphone for a year at a service kiosk alone was estimated at 6\\% of the GDP per capita in Kenya in 2013). Internet users are even fewer than smartphone owners. Smartphones are therefore only going to be found in (moderately) large number in the richest areas. If anything, this will help to differentiate these specific areas from other areas, hence making the model more effective rather than less. This is an interesting remark, so we have added this argument in the discussion (l.228-240).

As a side note, internet communications will, in principle, be visible in \\emph{xDR} which could be mobilised in future analyses. 

> P-Values are not presented in the evaluation of either the baseline or the proposed model. It would be good to see these, if possible.

Since our samples are quite large and the obtained r² are fairly above 0, all p-values are mechanically extremely small (<2.2e-16, which is the factory practical limit in R). We have written this information in the legend of table 1 (between l.168 and l.169).

> Although error bars are presented (by running the model 30 times with different random seeds), I would be interested in seeing more analysis around the sensitivity wrt. the random selection process.

The error bars have been replaced with standard boxes/whiskers to add information about the sensitivity with respect to the selection process. See new figure 3.

> Some of the constants chosen appear fairly arbitrary; for instance the five thresholds mentioned on L87 and the 1,000 inhabitant threshold mentioned on L73. Consider explaining how they were chosen.

The five thresholds are based on the shape of the distribution of daily activities between all pairs of towers (approximately an inverse power law of exponent 1.81). Also, defining what constitutes rural areas is still generally considered an open debate. The 1000 inhabitant threshold is indeed arbitrary, although it seems to have become somewhat of a "default" value for the UN and others (e.g. fao.org/3/a0310e/A0310E07.htm). We have added these two justifications (l.79-82 \\& l.97-98).

> A brief discussion on the type of data collected within the Senegal census may be relevant here. The authors claim that "an entire census can be estimated", however only population density and electricity consumption levels are estimated. This may be because the Senegalese census consists exclusively of population count, but this should be explained.

We acknowledge that our phrasing was quite misleading in this instance (changed l.246). For the record, the Senegalese census does encompass many more questions (such as the nature of the roof cover or the type of toilets), none of which appeared particularly suitable for predictions in our case.

> Several feature descriptors are used for the hierarchical clustering process -- however, these features are not directly used when modelling population density/electricity consumption. This (superficially) seems like a wasted opportunity, it may be worth explaining why?

All the features are used in table~S1. Combining the results obtained from different features did not noticeably improve the overall predictions (see response to the editor's comments), as is now explained in l.219-228. As a matter of fact, we found that the results based on activity curves were consistently better than those based on network features, although only by a small margin. We believe that enlightening the network aspects in the main text might still be useful as the relative performance of the two approaches could be reversed in another context (for example, as electrification rate grows, the nocturnal characteristics of the curves could disappear). This is now underlined immediately after the reference to table~S1 (l.191-193).

> A minor point, but the X-ticks for hour of day plots may be slightly more natural as [4, 8, 12, 16, 20]

Fixed. See new figures 1 and 4.

Reviewer #2's comments

General comments:

> My general comment is regarding mobile phone penetration and the frequency of usage of mobile phone service in developing countries. First of all, I just googled “the percentage of the world has a cell phone in 2019”, it shows 67 percent from statista.com. I do not know how reliable the data I found on google is. But I am interested in knowing whether this method can apply to other developing countries. However, based on the results of this paper, I speculate the mobile phone penetration rate is very high in Senegal. I appreciate the authors mention in the introduction that even in low electrification rural areas, mobile phone penetration in those areas are still 75%.

The method does require access to a sufficiently large mobile phone dataset and it is true that Senegal tends to be top of the class for sub-Saharan Africa in terms of electrification rate, mobile phone penetration and census data collection. That being said, the 67% figure is most likely an under-representation, since mobile phones can be shared in poor areas. Note also that we are not using the full penetration rate, but only the 65\\% market share of Sonatel. As a result, we believe that there should be enough underlying data in many other developing countries, and we are aware of some interested in this approach. We acknowledge that obtaining access to the data is however not straightforward since it is usually privately owned. We have included these arguments at the end of the conclusion (l.262-270).

> Secondly, mobile phone usage may vary across ages and/or education levels. People of different ages may have quite a different percentage of mobile usage even they all own a cell phone. In some extreme cases, children or school-age teenagers may not be encouraged to have/use a mobile phone. Then the lack of information from these categories of the population may affect the prediction power of the learning algorithm.

As mentioned above in the response to reviewer #1, we are only comparing towers among themselves and establishing similarity of usage. Hence, different usage inside the population that do not break similarity between places do not impact the results. Specifically, if the variable `age distribution inside an area' has a significant impact on phone usage then the clustering will group together places with similar age distributions, so that these will be represented by adequate reference towers. See text added l.228-240.

> Thirdly, regarding the representability of the mobile data from one provider. I appreciate the author uses the largest Senegalese telecommunication operator’s data, 65 percent of market share. From the results, I believe in Senegal, the other providers more or less target on similar categories of people compared to Sonatel. But in some cases (if extend this method to another developing country), different providers may target different categories of people. Some people choose to use a cheaper provider and they may choose to consume less amount of electricity, which may lead to bias in the model forecasting. The overall penetration of mobile phones and their frequency of usage in a particular developing country (like Senegal) may be introduced in the introduction. It would be interesting to know if in the case of low mobile penetration and/or high diverge in mobile phone usage in a developing country, how effective this method will be and what is the authors’ recommendation to use their method to uncover census data in the above cases. Besides those, I appreciate the authors take the consideration of tower in the data aggregation.

We have now emphasised the possible market share bias in a short discussion about possible extensions to other countries at the end of the conclusions (l.262-270), and have also added the market share information in the introduction (l.35). It is our belief that the only way to truly circumvent market share issues is probably not methodological, but rather by working on convincing different operators to release their data conjointly. The data we use for this specific project is already aggregated, so we cannot under-sample it to test low-penetration rates (and we would not be able to remove some targeted age or socio-economic groups anyway due to the information missing). We appreciate nonetheless this remark and keep it in mind in case some data allowing targeted under-sampling become available.

Minor comments:

> Page 1, line 17, What are the network characteristics you refer to?

Added examples l.18.

> Page 3 line 83, What is the definition of ‘distance matrices’ in your paper? Can you also give more details on the ‘point-by-point’ correlation you refer? What is the definition of ‘point’?

See improved version l.88-93.

> Page 5 line 161, What is the total number of towers? I agree that the authors remove 54 towers with no activity throughout the year in the clustering algorithm approach.

> Page 5 Table 1, In table 1, illustrations in the Voronoi cells around towers section (use average mobile data instead of the clustering algorithm), as a comparison, do you also remove the 54 inactive towers? If not, why? I speculate if the inactive towers are included, it will reduce the value of the correlation.

After verification, Table~1 was indeed computed with the inactive towers removed. The removal of the 54 inactive towers is now mentioned directly at the very beginning (l.50-51).

> Figure 3, Why are the correlations values of towers - calls(n) and towers-texts or towers-length(n) and towers- texts be the only ones that are selected as the horizontal lines? I can see that calls(n) are the highest correlation value in pop.all and length(n) is the highest one in elec.all (both of them are in terms of aggregating of towers).

These lines are in fact the maximum and minimum (at national level) to visualise the full range. Since the bottom line is not really necessary and confusing, we have removed it from the new fig.~3 and have updated the caption accordingly (between l.193-194).

> Supporting documents – S1\\_Appendix, Figure S1, It would be interesting if you can also add the performance from the “fully random samples of increasing size” (the grey lines in Figure 3).

Done.

> Minor typos I found from the manuscript:

Page3 line 83, Left quotation mark on the top left of the word “parallel”.

Page 4 Line 143, The extra word “calls” after “number of calls”.

Fixed.

Other

Completed reference [17], published during the review process.

Once again, we would like to thank the Editor and the Reviewers for their time.

Yours sincerely,

Hadrien Salat (corresponding author)

---

## [Decision Letter · Decision Letter 1]

8 Jun 2020

PONE-D-20-02661R1

A method to estimate population densities and electricity consumption from mobile phone data in developing countries

PLOS ONE

Dear Dr. Salat,

Thank you for submitting your manuscript to PLOS ONE. After careful consideration, we feel that it has merit but does not fully meet PLOS ONE’s publication criteria as it currently stands. Therefore, we invite you to submit a revised version of the manuscript that addresses the points raised during the review process.

Both reviewers are happy with your revision and recommended acceptance, but Reviewer 2 has a couple of additional minor comments on exposition. I also have one: in the abstract, "underwhelming" seems inappropriate, what does "a correlation is underwhelming" exactly mean? Please consider rephrasing this.

We look forward to receiving your revised manuscript.

Kind regards,

Shihe Fu, Ph.D.

Academic Editor

PLOS ONE

Reviewers' comments:

Reviewer's Responses to Questions

**Comments to the Author**

1. If the authors have adequately addressed your comments raised in a previous round of review and you feel that this manuscript is now acceptable for publication, you may indicate that here to bypass the “Comments to the Author” section, enter your conflict of interest statement in the “Confidential to Editor” section, and submit your "Accept" recommendation.

Reviewer #1: All comments have been addressed

Reviewer #2: All comments have been addressed

2. Is the manuscript technically sound, and do the data support the conclusions?

Reviewer #1: Yes

Reviewer #2: Yes

3. Has the statistical analysis been performed appropriately and rigorously? 

Reviewer #1: Yes

Reviewer #2: Yes

4. Have the authors made all data underlying the findings in their manuscript fully available?

Reviewer #1: Yes

Reviewer #2: Yes

5. Is the manuscript presented in an intelligible fashion and written in standard English?

Reviewer #1: Yes

Reviewer #2: Yes

6. Review Comments to the Author

Reviewer #1: The authors have suitably addressed all of my previous concerns/comments in their revised manuscript.

Reviewer #2: Thank you for the revision, which addresses issues I previously raised. This paper reflects scientific soundness. Therefore, I recommend acceptance.

Some minor questions to the author:

1. The author adds sentences "The map was created by the authors using R", what information authors would like to convey?

2. In the description of Figure 3, the best direct correlation from Table ?? for each case is represented..., which Table?

7. PLOS authors have the option to publish the peer review history of their article (what does this mean?). If published, this will include your full peer review and any attached files.

Reviewer #1: Yes: Joseph Redfern

Reviewer #2: No

---

## [Author Response · Author response to Decision Letter 1]

8 Jun 2020

Dear Dr. Fu,

We thank the Editor and the Reviewers for their new comments and positive feedback.

Editor's comment

> In the abstract, "underwhelming" seems inappropriate, what does "a correlation is underwhelming" exactly mean? Please consider rephrasing this.

We have changed "underwhelming" to "insufficiently high to provide an accurate representation of the situation" in the abstract.

Reviewer #2's comments

> 1. The author adds sentences "The map was created by the authors using R", what information authors would like to convey?

During re-submission, the in-house checks revealed that we needed to either "(1) present written permission from the copyright holder to publish [our] figures specifically under the CC BY 4.0 license, or (2) remove the figures from [our] submission". The quoted sentence was added to indicate that the maps do not, in fact, contain any copyrighted material as we created them ourselves. The justification within the submission system is probably enough, so we have removed the two clumsy sentences from the text at this stage.

> 2. In the description of Figure 3, the best direct correlation from Table ?? for each case is represented..., which Table?

Well spotted, thank you! This has now been fixed.

Other

We have changed the named contact at Sonatel to whom data inquires should be addressed, also requested during the in-house checks, as we are now aware that someone else within Sonatel is a better fit for the role.

We thank the Editor and the Reviewers for their new comments.

Yours sincerely,

Hadrien Salat

(corresponding author)

---

## [Editor Report · Decision Letter 2]

11 Jun 2020

A method to estimate population densities and electricity consumption from mobile phone data in developing countries

PONE-D-20-02661R2

Dear Dr. Salat,

We’re pleased to inform you that your manuscript has been judged scientifically suitable for publication and will be formally accepted for publication once it meets all outstanding technical requirements.

Kind regards,

Shihe Fu, Ph.D.

Academic Editor

PLOS ONE
---

## [Editor Report · Acceptance letter]

15 Jun 2020

PONE-D-20-02661R2 

A method to estimate population densities and electricity consumption from mobile phone data in developing countries 

Dear Dr. Salat:

I'm pleased to inform you that your manuscript has been deemed suitable for publication in PLOS ONE. Congratulations! Your manuscript is now with our production department. 

Kind regards, 

on behalf of

Dr. Shihe Fu 

Academic Editor

PLOS ONE